# Lung Carcinoids: A Comprehensive Review for Clinicians

**DOI:** 10.3390/cancers15225440

**Published:** 2023-11-16

**Authors:** Dan Granberg, Carl Christofer Juhlin, Henrik Falhammar, Elham Hedayati

**Affiliations:** 1Department of Molecular Medicine and Surgery, Karolinska Institutet, 17176 Stockholm, Sweden; henrik.falhammar@ki.se; 2Department of Breast, Endocrine Tumors and Sarcomas, Karolinska University Hospital Solna, 17176 Stockholm, Sweden; elham.hedayati@ki.se; 3Department of Oncology-Pathology, Karolinska Institutet, 17164 Stockholm, Sweden; christofer.juhlin@ki.se; 4Department of Pathology and Cancer Diagnostics, Karolinska University Hospital Solna, 17176 Stockholm, Sweden; 5Department of Endocrinology, Karolinska University Hospital Solna, 17176 Stockholm, Sweden

**Keywords:** lung carcinoids, symptoms, diagnosis, treatment, prognosis

## Abstract

**Simple Summary:**

Lung carcinoids are divided into typical and atypical. Most tumors are slow-growing yet have malignant potential, which is more common in patients harboring atypical carcinoids. A large proportion of these patients are diagnosed incidentally on chest X-ray or CT scan. Cough, dyspnea, or recurrent pneumonia are common presenting symptoms. Endocrine symptoms, such as carcinoid syndrome or ectopic Cushing’s syndrome, are uncommon. Most individuals are cured by surgery, but some tumors metastasize. For patients with metastatic disease, chemotherapy, peptide receptor radionuclide therapy (PRRT), or targeted therapies are alternatives. In this article, we review the pathology, symptoms, diagnosis, and treatment of patients with lung carcinoids.

**Abstract:**

Lung carcinoids are neuroendocrine tumors, categorized as typical or atypical carcinoids based on their histological appearance. While most of these tumors are slow-growing neoplasms, they still possess malignant potential. Many patients are diagnosed incidentally on chest X-rays or CT scans. Presenting symptoms include cough, hemoptysis, wheezing, dyspnea, and recurrent pneumonia. Endocrine symptoms, such as carcinoid syndrome or ectopic Cushing’s syndrome, are rare. Surgery is the primary treatment and should be considered in all patients with localized disease, even when thoracic lymph node metastases are present. Patients with distant metastases may be treated with somatostatin analogues, chemotherapy, preferably temozolomide-based, mTOR inhibitors, or peptide receptor radionuclide therapy (PRRT) with ^177^Lu-DOTATATE. Most patients have an excellent prognosis. Poor prognostic factors include atypical histology and lymph node metastases at diagnosis. Long-term follow-up is mandatory since metastases may occur late.

## 1. Introduction

Neuroendocrine tumors may arise in many organs in the body, including the gastrointestinal tract, lungs, and sympathetic and parasympathetic ganglia. The presence of neuroendocrine cells in the lungs was first described by Frölich in 1949 [1] and confirmed by Feyrter in 1954 [2]. These cells may be solitary or occur in clusters, termed neuroepithelial bodies [3,4]. Neuroendocrine cells in the lungs are believed to be the origin of neuroendocrine lung tumors [5,6,7,8]. Neuroendocrine lung tumors are divided into typical and atypical carcinoids, large cell neuroendocrine carcinomas, and small cell lung carcinomas. Arrigoni et al. were the first to introduce the division of pulmonary neuroendocrine tumors into typical and atypical carcinoids and small-cell lung carcinomas [9]. More recently, several other classifications have been suggested. For a long time, lung carcinoids were considered relatively benign neoplasms, and patients were often monitored with chest X-rays for a few years before being discharged. Although both typical and atypical carcinoids are generally slow-growing tumors, it is now understood that certain tumors indeed possess malignant potential, capable of metastasizing even many years after curative-intent surgery. This article reviews the pathology, symptoms, diagnosis, treatment, and follow-up of patients with typical and atypical carcinoids. In addition, we briefly discuss the conditions of diffuse idiopathic neuroendocrine cell hyperplasia (DIPNECH) and lung carcinoid tumorlets.

## 2. Epidemiology and Etiology

The incidence of lung carcinoids has risen over the past few decades. Currently, it stands at approximately 0.7 per 100,000 among Caucasians and 0.5 per 100,000 among Black individuals. Women are slightly more affected than men [10,11,12]. The disease occurs in all ages, even in children, yet is most frequent in middle age [10,12]. The etiology is unknown; no relation to smoking has been found for typical carcinoids, yet smoking may increase the risk for atypical carcinoids [13,14]. The risk of lung carcinoids is increased in patients with multiple endocrine neoplasia type 1 (MEN1), seen in 5–35% of those patients, more commonly in males and smokers. The tumors in MEN1 patients are usually small, multifocal, located peripherally in the lungs, and recur frequently [15]. Typical carcinoids account for 2% and atypical carcinoids for 0.2% of all lung tumors; adenocarcinomas constitute 40%, squamous cell carcinomas 30%, and small cell lung carcinomas 15% of all lung cancers [16].

## 3. Clinical Presentation

Up to half of the patients (13–51%) with lung carcinoids have no symptoms and are incidentally diagnosed on routine chest X-rays or computerized tomography (CT) scans. Common presenting symptoms include classic pulmonary symptoms such as dry cough, wheezing sound, hemoptysis, dyspnea, recurrent pneumonia, persisting lung infiltrates, and chest pain [17,18,19,20]. Delayed diagnosis, sometimes for several years, may occur due to misdiagnosis as asthma. Although lung carcinoids may secrete various hormones, endocrine symptoms are rare. Carcinoid syndrome, due to elevated 5-hydroxy indoleacetic acid (5-HIAA) with flushes, diarrhea, asthma, and right-sided valvular heart disease, is normally seen only when liver metastases are present and is seen in 2–12% of patients with lung carcinoids [21,22,23]. The low frequency of carcinoid syndrome can be explained by the high concentration level of monoamine oxidase in the pulmonary system, which metabolizes serotonin, and the rare occurrence of distant metastases in patients with lung carcinoids. An atypical carcinoid syndrome with generalized flushing, edema, lacrimation, bronchoconstriction, and diarrhea caused by histamine secretion may be seen occasionally. About 2–6% of patients suffer from ectopic Cushing’s syndrome due to secretion of corticotropin-releasing factor or adrenocorticotropic hormone (ACTH) [22,24]. Production of growth-hormone-releasing hormone causing acromegaly is rare [25,26].

Neuroendocrine cell hyperplasia may be found in patients with chronic lung diseases, including bronchiectasis and fibrosis, or those who are heavy smokers. When there are no predisposing conditions, the entity is called DIPNECH, which is most common in middle-aged women. Patients with DIPNECH may experience cough, dyspnea, and wheezing. Both tumorlets and DIPNECH may occur concomitantly with lung carcinoids.

Metastatic disease occurs in 5–20% of patients with typical carcinoids and up to 70% of patients with atypical carcinoids. Metastases are most frequently seen in regional lymph nodes but may also arise distantly in the liver, bones, brain, subcutaneous tissue, mammary glands, eyes, and adrenals [9,21,27,28]. Metastases may occur late, even decades, after primary tumor surgery [29].

## 4. Diagnosis

### 4.1. Macroscopic Pathology

A majority (60–84%) of lung carcinoids are centrally located in the main or lobar bronchi and may be detected at bronchoscopy as a polypoid, highly vascular, intrabronchial tumor, often infiltrating the surrounding lung parenchyma [30]. Peripheral carcinoids are not accessible by bronchoscopy. Lung carcinoids may be multiple and surrounded by tumorlets and/or DIPNECH. Atypical carcinoids are often peripheral, while typical carcinoids may be situated anywhere in the lungs.

### 4.2. Histopathology

According to the WHO classification from 1999 [31], lung carcinoids are divided into typical and atypical carcinoids [32], based on Travis et al. [33] and revised in 2004. The division into typical carcinoids or atypical carcinoids is based on the number of mitoses per 2 mm^2^ and the presence of necrosis in histology. Those tumors with less than two mitoses per 2 mm^2^ and without necrosis are relatively benign and are referred to as typical carcinoids. Those tumors with 2–10 mitoses per 2 mm^2^ and/or necrosis have a higher malignant potential and are referred to as atypical carcinoids [34]. A revision performed in 2015 grouped all neuroendocrine lung tumors, including typical and atypical carcinoids, large cell neuroendocrine carcinomas, and small cell lung carcinomas, into one neuroendocrine lung tumor group. The revision included specified methods for counting mitoses, in which the areas of highest activity should be used for counting [35]. The current recommendation does not advise using the Ki67 proliferative index as a means to differentiate between typical and atypical carcinoids, setting them apart from most other gradable neuroendocrine tumors.

The histological hallmarks of carcinoids encompass neuroendocrine differentiation, featuring neuroendocrine growth patterns, a “salt and pepper” chromatin pattern, inconspicuous nucleoli, and moderate to abundant cytoplasm. Typical carcinoids are usually composed of small polyhedral cells with small nuclei that are round or oval. They have eosinophilic, finely granular cytoplasm and are arranged in various growth patterns such as trabecular, insular, palisading, ribbon-like, or rosette-like structures, all separated by a fibrovascular stroma. Nuclear molding is not present. Mitotic activity is infrequent, and necrosis is not observed (Figure 1). Atypical carcinoids feature increased mitotic counts and/or punctate necrosis but are otherwise quite similar to typical carcinoids in terms of cytomorphology and growth patterns. Occasionally, nuclear pleomorphism is noted (Figure 2).

### 4.3. Diffuse Idiopathic Neuroendocrine Cell Hyperplasia (DIPNECH) and Carcinoid Tumorlets

Proliferation of pulmonary neuroendocrine cells may remain confined to the respiratory mucosa (DIPNECH) or progress locally, leading to the development of tumorlets and carcinoid tumors. In contrast, the tumor cells in tumorlets extend beyond the respiratory epithelial basement membrane [34]. Tumorlets show the same morphology as typical carcinoids but are not larger than 5 mm in size.

### 4.4. Immunohistochemistry

From an immunohistochemical viewpoint, lung carcinoids are positive for pan-cytokeratins and neuroendocrine markers of first- (CD56, chromogranin A, and synaptophysin) and second-generation lineages (INSM1). Several hormones, including serotonin (84%), pancreatic polypeptide, gastrin, gastrin-releasing peptide, calcitonin, ACTH, and growth-hormone-releasing hormone often show positive immunostaining in lung carcinoids; multiple hormones may be found in the same tumor [19,36]. S-100 protein expression is most common in peripheral tumors [37,38]. The adhesion molecule CD44 has prognostic implications in typical carcinoids and is expressed in most lung carcinoids [39]. The retinoblastoma gene protein also frequently shows positive immunohistochemistry in typical carcinoids [40,41], while the proteins *p53* and *BCL-2* are usually negative, yet more often positive in atypical carcinoids [42,43]. The proliferation marker Ki67 is generally low in typical carcinoids but may be higher in atypical tumors [44]. Thyroid transcription factor-1 (TTF-1) stains positive in 28–69% of lung carcinoids [45,46] and may help in the differential diagnosis between a primary lung carcinoid and metastasis from a neuroendocrine tumor located elsewhere. The marker is considered highly specific (but not sensitive) for pulmonary neuroendocrine neoplasms [46]. Staining for bombesin may be useful to support a pulmonary origin in terms of metastatic spread in cases without a known primary tumor [47].

### 4.5. Genetic Alterations

Deletions in the *MEN1* locus at 11q [48] are common in both typical and atypical carcinoids. Homozygous somatic inactivation of the *MEN1* gene has been reported in 36% of sporadic lung carcinoids [49]. Deletions of chromosome 10q or 13q are frequent in atypical carcinoids [48]. Aneuploidy has been detected and reported in 5–32% of typical carcinoids as well as in 17–79% of atypical carcinoids [50,51,52,53]. Subsets of cases may harbor mutations in clinically relevant genes such as *EGFR* and *PIK3CA*, for which targeted therapy may be an option [54].

### 4.6. Radiology

The tumor is visible on chest X-rays in more than 60% of patients [24,55]. Patients with central tumors may have peripheral atelectasis or pneumonic infiltrates as signs of bronchial obstruction. CT scan is more sensitive than chest X-ray [56,57,58] and should always be performed to detect enlarged lymph nodes, tumorlets, and DIPNECH and delineate the tumor. Magnetic resonance imaging (MRI) scan is less sensitive than CT scan to detect small lung lesions.

Somatostatin receptor scintigraphy may visualize about 70–80% of lung carcinoids [59]. Positron emission tomography (PET) with ^68^Ga-DOTATOC or ^68^Ga-DOTATATE is more sensitive than somatostatin receptor scintigraphy to detect somatostatin-receptor-positive neuroendocrine tumors [60] (Figure 3). ^68^Ga-PET/CT is recommended preoperatively to determine whether it can be used for postoperative follow-up, as well as for staging of the disease [61]. PET with ^68^Ga is especially valuable in patients with Cushing’s syndrome who often have small tumors, with up to 12 years delay in localizing the tumor [62]. The benefit of PET with ^18^F-fluorodeoxyglucose (FDG) is more controversial since these tumors often have lower uptake than expected for malignant tumors (Figure 1) [63,64]. However, in a study of 16 patients with lung carcinoids (11 typical, 5 atypical) the overall detection sensitivity was 75%. The authors concluded that PET with FDG is valuable for evaluating typical and atypical lung carcinoids [65].

Bronchoscopy is performed in most patients and can identify central intrabronchial tumors (Figure 4).

Since brushing or sputum cytology is often negative, it is crucial to take biopsies to obtain a correct preoperative diagnosis. It is generally considered safe to take biopsies despite the risk of bleeding [66]. Peripheral tumors may be biopsied transthoracically guided by CT, although misdiagnosis as small cell lung carcinoma is not uncommon. Staining for Ki67 may aid in the differential diagnosis between atypical carcinoid and small cell lung carcinoma. A Ki67-index > 25% favors the diagnosis of small cell lung cancer, while a lower proliferative rate indicates carcinoid [67,68].

Plasma chromogranin A is usually at normal concentrations. Preoperative elevation of chromogranin A should warrant an intense search for distant metastases. Analysis of 5-HIAA, cortisol, or ACTH is not indicated unless endocrine symptoms are present.

### 4.7. Staging

Staging of lung carcinoids is performed after TNM (tumor, nodes, metastases) classification according to the eighth edition of lung cancer stage classification [69].

### 4.8. Differential Diagnoses

Differential diagnoses of a radiologically identified lung tumor include other lung neoplasms, hamartoma, and metastasis from another primary tumor. The histopathological differentiation between atypical carcinoid and small cell lung cancer may sometimes be difficult but is clinically significant since patients with carcinoids are most often cured by surgery. In contrast, patients with small cell lung cancer are treated with chemoradiotherapy.

## 5. Treatment

Most patients with lung carcinoids are diagnosed before the occurrence of distant metastases. Surgery is curative in most of these cases. In individuals with distant metastases, the treatment is more controversial. A watch-and-wait policy has been proposed in asymptomatic individuals with a low proliferative rate [61]. Possible treatment options include somatostatin analogues, chemotherapy, mTOR inhibitors, and radionuclide therapy with ^177^Lu-DOTA-octreotate, depending on somatostatin receptor expression, proliferative rate, bone marrow and kidney function, and the patient’s general health. A summary of treatments for individuals with metastatic disease is given in Table 1.

### 5.1. Surgery

For patients with typical and atypical lung carcinoids without distant metastases, surgery involving the complete removal of the primary tumor and a systematic lymph node dissection, which eliminates all affected lymph nodes, is recommended as the sole curative treatment. It is essential to preserve as much healthy lung parenchyma as possible. Surgical methods include bronchotomy with excision of the tumor and bronchoplasty, sleeve resection where the part of the bronchus containing the tumor is resected and an end-to-end anastomosis is performed, wedge or segmental resection, lobectomy, bilobectomy, and pneumonectomy. When deciding the surgical procedure, the type of tumor (typical/atypical), presence of lymph node metastases and surrounding tumorlets, and the age and lung function of the patient must all be taken into account. In contrast to lung cancer, carcinoids do not require wide resection margins. Peripheral tumors can be removed by wedge resection, segmentectomy, or lobectomy/bilobectomy. Small central tumors can be treated with bronchotomy with resection of the tumor and bronchoplasty, sleeve resection of the bronchus, or segmentectomy [84,85,86], while larger central tumors may require lobectomy or bilobectomy. An advantage of sleeve resection is that the hospital morbidity and mortality are very low [84] and more lung function is preserved. Pneumonectomy is reserved for large invasive central tumors. In older patients and patients with reduced lung function, great efforts should be made to preserve maximum lung parenchyma. Intra-operative biopsies with examination of frozen sections are warranted. Individuals diagnosed with atypical carcinoids should undergo at least a lobectomy as a basic surgical intervention [58,61,87]. Nowadays, most surgeries for lung carcinoids are performed by video-assisted thoracoscopic surgery (VATS) [88], which leads to better quality of life as well as better-preserved lung and shoulder function compared with open surgery [89]. Even if there is a small risk for conversion to open surgery, especially in larger tumors, this does not result in more postoperative complications [90].

### 5.2. Interventional Pulmonology

Since lung carcinoids often grow profoundly into the surrounding tissue, endoscopic excision of the mass by YAG laser is usually not recommended. Two studies, however, found that bronchoscopic laser therapy was a safe and effective treatment option for approximately 54–64% of individuals with typical intrabronchial carcinoids; open surgery had to be performed later for the remaining patients [91,92]. If possible, a sleeve resection may instead be performed to avoid a lobectomy and repeated surgery [85]. Open surgery is not recommended for individuals with a high risk of cardiopulmonary complications. Instead, their obstructive symptoms may be palliated by YAG-laser-mediated removal or reduction of an intrabronchial carcinoid. Moreover, in a few patients, laser treatment to reduce the tumor mass may enable surgery after post-obstructive infiltrates have resolved. There is no scientific evidence for adjuvant chemotherapy after radical surgery [93,94].

### 5.3. Radiotherapy

External radiotherapy is mainly performed to alleviate pain from bone metastases and treat brain metastases but is also recommended as a complementary treatment modality after incomplete resection or palliative treatment in inoperable tumors. Peptide receptor radionuclide therapy (PRRT) is a possibility in individuals with metastatic or inoperable tumors showing high expression of somatostatin receptors on ^68^Ga-DOTATOC or ^68^Ga-DOTATATE PET. In one small study, a radiological response was observed among 5/9 patients receiving ^177^Lu-DOTA-octreotate for a median of 31 months [75], and in another study, 29% (24/84) treated with ^90^Y-DOTA-octreotide responded objectively [76]. Since 9.5% of all patients treated with the ^90^Y-DOTA-octreotide experienced a serious permanent worsening of the kidney function, ^177^Lu-DOTA-octreotate may be preferable. Ianniello et al. treated 34 patients having progressive lung carcinoids (15 typical, 19 atypical) with four to five cycles of ^177^Lu-DOTA-octreotate up to a cumulated activity of 18.5 or 27.8 GBq. Individuals with typical carcinoids achieved a disease control rate of 80% (6% complete response, 27% partial response, and 47% stable disease) with a median PFS of 20.1 months, while individuals with atypical carcinoids had a disease control rate of 47.5% (no objective response) with a median PFS of 15.7 months [77]. Sabet et al. reported 22 patients that received four cycles of 7.8 GBq ^177^Lu-DOTATATE with three-month intervals; partial response occurred in 6 (27.3%) and stable disease in 9 (40.9%). Median PFS was 27 months, and median overall survival (OS) was 42 months [78]. Another, more extensive, retrospective study encompassing 114 patients compared three protocols: ^90^Y-DOTATOC, ^177^Lu-DOTATATE, and ^90^Y-DOTATOC + ^177^Lu-DOTATATE. Median OS was 58.8 months, and median progression-free survival (PFS) was 28.0 months. In that study, 15 patients (13.3%) had objective response and 61 (53.5%; including 15 individuals with minor response) had stable disease. Individuals treated with the combination of ^90^Y-DOTATOC and ^177^Lu-DOTATATE showed the best objective response rate (38.1%) [79].

### 5.4. Somatostatin Analogues

The somatostatin analogues octreotide and lanreotide have previously demonstrated antitumoral activity in gastrointestinal and pancreatic neuroendocrine tumors [95,96]. Sullivan et al. evaluated 61 patients with lung carcinoids (20 typical, 41 atypical) receiving octreotide long-acting release (LAR) 20 or 30 mg intramuscularly every four weeks or lanreotide LAR 90 or 120 mg subcutaneously every four weeks. Forty-one patients were slowly progressing before the start of somatostatin analogue treatment. The best response was stable disease seen in 47 patients (77%). Median PFS and median OS were 17.4 months and 58.4 months, respectively. Individuals with slowly progressive disease before the somatostatin analogue and patients with functioning tumors had significantly longer PFS [70]. In a more recent report, 31 consecutive individuals, 14 with typical and 17 with atypical lung carcinoids, used first-line octreotide LAR or lanreotide depot every four weeks. A majority (60%) had Ki67 ≤ 10%. Partial response was observed in 2 individuals (6.5%), stable disease in 24 (77.4%), and progressive disease in 5 (16.1%) individuals. Median PFS was 28.6 months, and median OS was not achieved. Median PFS was longer, yet not significantly, in individuals with typical carcinoids and Ki67 ≤ 10% [71]. The double-blind SPINET trial, randomizing individuals with somatostatin-receptor-positive typical and atypical carcinoids 2:1 to lanreotide LAR 120 mg s.c. or placebo every four weeks, was ended early due to slow accrual after inclusion of 77 individuals (52 lanreotide, 26 placebo). Results were presented at the 2021 ESMO meeting. PFS was 21.9 months for lanreotide vs 13.9 months for placebo in individuals with typical carcinoids and 13.8 months for lanreotide vs 11.0 months for placebo in atypical carcinoids. The authors concluded that lanreotide 120 mg every 4 weeks could be an appropriate treatment especially for individuals with typical carcinoids [97].

### 5.5. Chemotherapy

In individuals with metastatic lung carcinoids, several chemotherapy regimens have shown to have limited response rates. The modalities that have been studied are single cisplatin and docetaxel, carboplatin + etoposide, paclitaxel ± doxorubicin, streptozotocin + 5-fluorouracil or doxorubicin, oxaliplatin + capecitabine, 5-fluorouracil + dacarbazine + epirubicin, and 5-fluorouracil + cisplatin + streptozotocin [98,99,100,101,102,103,104]. The best results were observed with temozolomide. In one report encompassing 31 individuals with typical or atypical carcinoids, objective tumor response was demonstrated in 14% and stabilization of progressive disease in 52%. All individuals with partial response were found to have atypical carcinoids, but stabilization was noted in both typical and atypical carcinoids [72]. Combining temozolomide with capecitabine has yielded similar results. Papaxoinis et al. treated 33 individuals with well-differentiated lung carcinoids (10 typical, 20 atypical, 3 not specified) with capecitabine 750 mg/m^2^ twice daily day 1–14 and temozolomide 200 mg/m^2^ day 10–14, repeated every four weeks for at most six cycles, followed by maintenance therapy with octreotide LAR 30 mg intramuscularly every four weeks. Partial response was found in 6 (18%) and stable disease in 19 (58%) patients. The median response duration was 21.7 months, median PFS was 9.0 months, and median OS was 30.4 months [74].

### 5.6. Targeted Therapies

Several medications targeting signal pathways or membrane receptors have shown activity in individuals with neuroendocrine neoplasia. In 2011, two reports were published demonstrating that both everolimus, an inhibitor of mammalian target of rapamycin (mTOR), and sunitinib, inhibiting vascular endothelial growth factor receptor (VEGFR), platelet-derived growth factor receptor (PDGFR), and c-kit, prolong PFS in persons with pancreatic endocrine tumors [105,106].

### 5.7. mTOR Inhibitors

Everolimus was also studied in individuals with lung carcinoids. In a subanalysis from the randomized, placebo-controlled RADIANT-2 study, PFS in lung carcinoid patients treated with everolimus + octreotide LAR was 13.6 months vs 5.6 months in the group with placebo + octreotide LAR. However, no significant difference was found. A small response was observed in 67% and 27% in the everolimus and placebo groups, respectively [80]. In another subanalysis from the randomized phase 3 RADIANT-4 study, encompassing 90 individuals with well-differentiated lung carcinoids, everolimus 10 mg od (*n* = 63) was compared with placebo (*n* = 27). Median PFS was 9.2 months (95% CI 6.8–10.9) in the everolimus group and 3.6 months (95% CI 1.9–5.1) in the placebo group. The risk for disease progression or death was reduced by 50% in the everolimus group. Tumor shrinkage was observed in 58% of the individuals receiving everolimus vs 13% among those that were on placebo [81]. In the LUNA study, everolimus (*n* = 42) was compared with long-acting pasireotide (*n* = 41) and everolimus + pasireotide (*n* = 41) in persons with well-differentiated lung and thymic carcinoids. Disease control rate (complete response + partial response + stable disease) after 9 months was 33.3% in the everolimus group, 39.0% in the pasireotide group, and 58.5% in the combination group [73]. These data suggest that everolimus has antitumor activity in individuals with lung carcinoids. According to recommendations in the Commonwealth Neuroendocrine Tumor Research Collaboration and the North American Neuroendocrine Tumor Society Guidelines, everolimus should be considered in progressing lung NETs, both non-functional and functional [107].

### 5.8. Anti-Angiogenic Drugs

The tyrosine kinase receptors PDGFRα, PDGFRβ, c-kit, and EGFR are expressed in most lung carcinoids [108]. However, neither sunitinib nor pazopanib, an inhibitor of VEGFR-1, VEGFR-2, VEGFR-3, PDGFRα, PDGFRβ, and c-kit, has been studied in lung carcinoids except as small parts of more extensive studies, with no convincing results [109,110].

### 5.9. Immune Therapy

Immune checkpoint inhibitor antibodies target the interaction between programmed death receptor 1 (PD-1) and its ligand PD-L1. Examples are pembrolizumab, nivolumab and atezolizumab. Antitumoral activity has been shown in several tumor types. There are few clinical data for patients with lung carcinoids. Spartalizumab, a humanized anti-PD-1 antibody blocking PD-L1 and PD-L2, was investigated in a phase 2 study with advanced non-functioning neuroendocrine tumors progressing on prior therapy. In 30 patients with thoracic NET, partial response was observed in 5 (16.7%), all harboring atypical tumors. In addition, 17 (56.7%) had stable disease [82]. In the KEYNOTE-028 study, individuals with advanced PD-L1-positive carcinoid or pancreatic neuroendocrine neoplasia were treated with pembrolizumab 10 mg/kg every fortnight for up to 2 years. Nine patients with lung carcinoids were included, of whom one had a partial response lasting seven months [83]. Although Tsuruoka et al. found no expression of PD-L1 in typical and atypical carcinoids [111], further testing of immune checkpoint inhibitors in persons with lung carcinoids would be highly interesting since antitumoral activity has been observed in other tumor types regardless of PD-L1 expression [112].

### 5.10. Local Treatment

Patients with progressing liver metastases and stable or no disease outside the liver or uncontrollable hormonal symptoms from liver metastases may benefit from debulking of the liver metastases. This may be performed by embolization of the hepatic arteries with particles causing ischemia in the metastases, ^90^Y-labeled microspheres (SIR-Spheres^®^ or TheraSpheres^®^) causing a local radiation effect in the metastases, or chemotherapeutic agents. Although radiofrequency or microwave ablation of liver metastases has not been shown to prolong survival in individuals with liver metastases from ileal neuroendocrine tumors [113], this can be an alternative for individuals with a limited number of liver metastases or patients in whom only one liver metastasis is progressing and the other remains stable.

### 5.11. Adjuvant Treatment

There are no data supporting prolonged survival with adjuvant chemotherapy or radiotherapy after surgical resection either of typical or atypical carcinoids [94,114,115].

### 5.12. Symptomatic Treatment

The primary symptomatic treatment for classical carcinoid syndrome is somatostatin analogues [61,116]. Patients with diarrhea and high 5-HIAA levels may in addition benefit from telotristate ethyl [117,118]. Bilateral adrenalectomy is the most efficient symptomatic therapy in persons with ectopic Cushing syndrome and metastatic disease [119]. Other options include ketoconazole, metyrapone [116], somatostatin analogues [120], and mitotane [119], which can also be used to correct the metabolic disturbances before surgery in patients without distant metastases.

### 5.13. Treatment of DIPNECH

There is no accepted treatment for patients with DIPNECH, but bronchodilators or inhaled corticosteroids could be tried in addition to surgical excision of the largest lesion(s). Systemic therapy with somatostatin analogues was shown to stabilize the condition [121]. DIPNECH is stable in approximately half of individuals, and the remaining half will develop progressive disease [122,123].

## 6. Prognosis

Most patients treated with surgical resection of the tumor will be cured. Five- and ten-year survival is 87–100% and 82–95%, respectively, for patients with typical carcinoids, and 40–93% and 31–67%, respectively, for patients with atypical carcinoids [19,24,124,125,126]. Poor prognostic factors include atypical histology, lymph node metastases at diagnosis, advanced stage, and presence of tumorlets [124,127,128,129,130]. In addition, one study found that patients with persistent Cushing syndrome after treatment had worse prognosis [131]. Regarding the Ki67 index, conflicting results have been obtained [44,127,128,132]. Apart from typical histology and absence of lymph node metastases, positive prognostic factors include positive immunostaining for CD44, the adhesion molecule, and positive nuclear staining for the metastasis suppressor gene *nm23* [39,132].

## 7. Follow-Up

Lung carcinoids may recur late, many years and even decades after primary surgery [29,133,134,135,136]. In order to detect late disease progression and recurrences after primary surgery, long-term follow-up is necessary. Some authors recommend life-long follow-up [116]. The radiological follow-up should include a low-dose thoracic CT scan and MRI scan of the abdomen. PET with ^68^Ga-DOTATOC/DOTATATE may be used in patients with somatostatin-receptor-positive tumors in case of equivocal radiology. Bronchoscopy examination needs to be repeated in a selected patient category [116]. The frequency of the radiological examinations has to be determined on an individual basis. Special attention is needed among patients with high proliferative rate, lymph node metastases at surgery, and atypical carcinoids. In patients with radically resected node-negative typical carcinoids, the risk of recurrence is very low, about 2%; it may be possible to abstain from follow-up in this patient group [135].

## 8. Conclusions

Most lung carcinoids are slow-growing neoplasms detected before distant metastases are present. Radical surgery, including resection of the primary tumor and affected lymph nodes, is curative in most cases. Adjuvant postoperative chemotherapy is not indicated. Patients with metastatic disease may be treated with chemotherapy (preferably temozolomide-based), somatostatin analogues, mTOR inhibitors, or PRRT with ^177^Lu-DOTATATE. Disease recurrence may occur late; hence, long-term follow-up is essential, except maybe in patients with radically operated typical carcinoids without lymph node metastases. Most patients have an excellent prognosis.

## Figures and Tables

**Figure 1 cancers-15-05440-f001:**
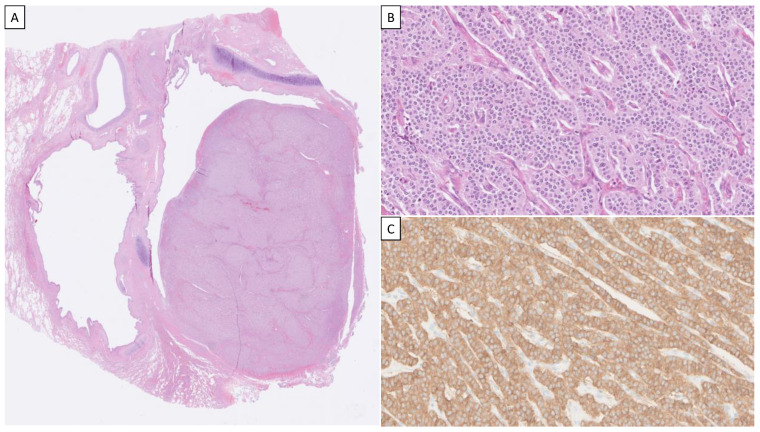
Histological and immunohistochemical hallmarks of a typical pulmonary carcinoid. (**A**) Low-power (×40 magnification) overview of a hematoxylin–eosin-stained slide depicting an endobronchial tumorous mass. (**B**) High-power magnification (×200) of tumor cells in cord-like arrangements. Note the lack of nuclear atypia, mitotic figures, and necrosis. (**C**) Typical carcinoids are diffusely positive for chromogranin A and synaptophysin; the latter staining is depicted here.

**Figure 2 cancers-15-05440-f002:**
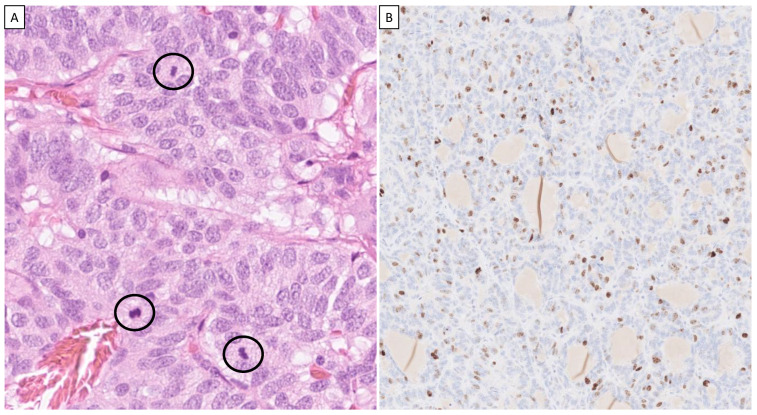
Atypical carcinoid. (**A**) Hematoxylin–eosin-stained tumor cells at ×400 magnification with slightly irregular tumor nuclei; several mitotic figures are noted (in circles). (**B**) The Ki-67 index was 15% (×200 magnification). Although not diagnostic, the Ki-67 index is usually much higher in atypical carcinoids compared with typical ones.

**Figure 3 cancers-15-05440-f003:**
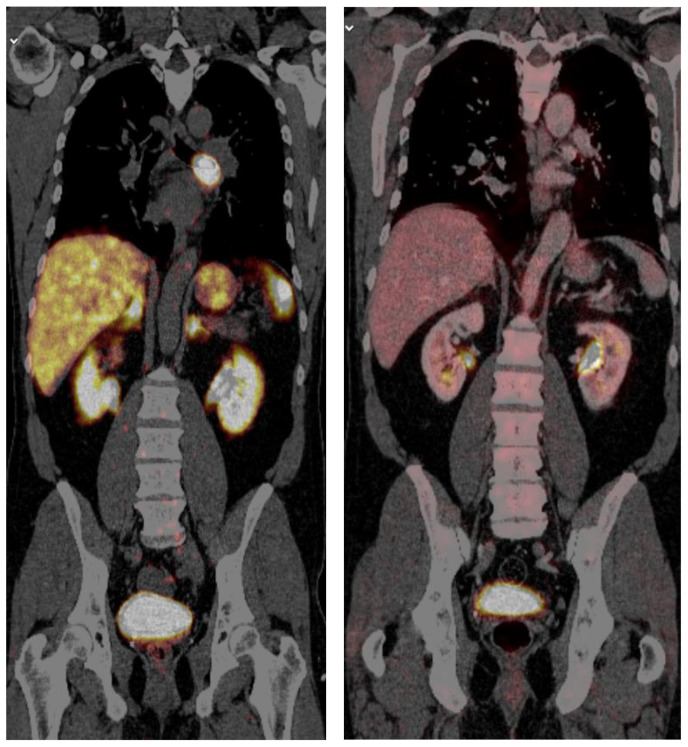
Carcinoid in the upper left lobe strongly positive on ^68^Ga-DOTATOC PET/CT (**left**) and negative on FDG PET/CT (**right**).

**Figure 4 cancers-15-05440-f004:**
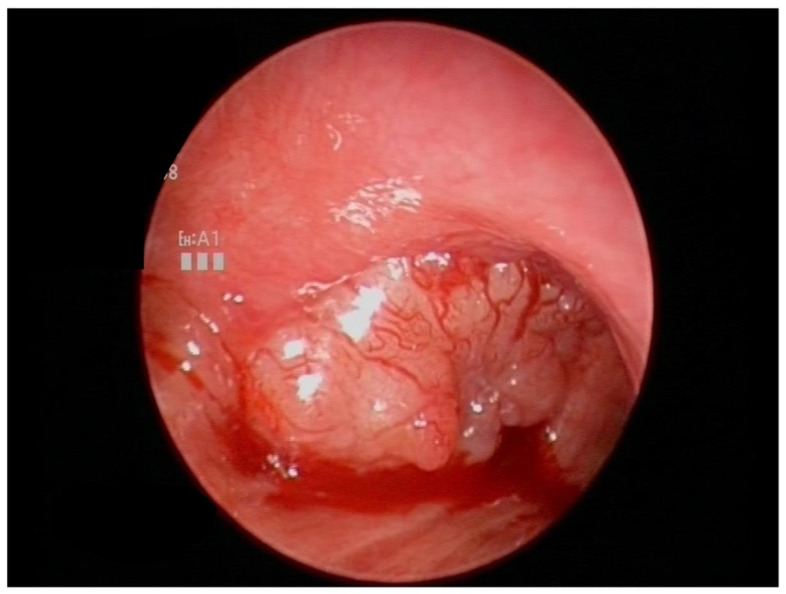
Carcinoid filling almost the whole right main bronchus.

**Table 1 cancers-15-05440-t001:** Therapies for patients with lung carcinoids.

	*n*	CR + PR	SD	PFS (mo)	OS (mo)	Reference
Octr, Lan	61	0	47 (77%)	17.4	58.4	[70]
Octr, Lan	31	2 (6.5%)	24 (77.4%)	28.6	NR	[71]
Tem	31	3 (14%)	11 (52%)	5.3	23.2	[72]
Pas	41	1 (2.4%)	14 (34.1%)	8.5	-	[73]
Cap + Tem	33	6 (18%)	19 (58%)	9.0	30.4	[74]
^177^Lu	9	5 (56%)	-	-	-	[75]
^90^Y	84	24 (29%)	-	-	40	[76]
^177^Lu	34	5 (15%)	16 (47%)	18.5	48.6	[77]
^177^Lu	22	6 (27.3%)	9 (40.9%)	27	42	[78]
^90^Y, ^177^Lu, ^90^Y + ^177^Lu	114	15 (13.3%)	61 (53.5%)	28.0	58.8	[79]
Everolimus + Octr	33	20 (66.7%) *	-	13.6	-	[80]
Everolimus	63	1 (1.6%)	50 (79.4%)	9.2	-	[81]
Everolimus	42	1 (2.4%)	13 (31.0%)	12.5	-	[73]
Everolimus + Pas	41	1 (2.4%)	20 (48.8%)	11.8	-	[73]
Spartalizumab	30	5 (16.7%)	17 (56.7%)	-	-	[82]
Pembrolizumab	9	1 (11.1%)	-	-	-	[83]

Octr, octreotide; Lan, lanreotide; Pas, pasireotide; Cap, capecitabine; Tem, temozolomide; ^177^Lu, ^177^Lu-DOTATATE; ^90^Y, ^90^Y-DOTATOC; NR, not reached; *, minor response.

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
