# Peer review of "Lung Carcinoids: A Comprehensive Review for Clinicians"

_cancers, 2023, doi:10.3390/cancers15225440_

Round 1

Reviewer 1 Report

Comments and Suggestions for Authors

Good overview on lung carcinoids

I have only minor comments

In the abstract, I would point out in the initial sentence that carcinoids are neuroendrocrine tumors, and clarify the abbreviation PRRT.

In the introduction, the entities of neuroendocrine tumors of the lungs (including small cell lung cancer and large cell neuroendocrine carcinoma) should be mentioned.

I find a little complicated to follow the flow of the manuscript. Why “Pathology” section is separated from “Diagnosis”? I understand that Pathology has a prominent role, but its main role is in allowing the diagnosis of these entities.

Authors should mention Staging (the procedures for staging are approached in the Diagnosis and Treatment sections).

Author Response

Response to Reviewer 1:

  1. In the abstract, I would point out in the initial sentence that carcinoids are neuroendrocrine tumors, and clarify the abbreviation PRRT.

Answer: The abstract has been changed according to the recommendation

  1. In the introduction, the entities of neuroendocrine tumors of the lungs (including small cell lung cancer and large cell neuroendocrine carcinoma) should be mentioned.

Answer: We have now mentioned these two entities in the introduction.

  1. I find a little complicated to follow the flow of the manuscript. Why “Pathology” section is separated from “Diagnosis”? I understand that Pathology has a prominent role, but its main role is in allowing the diagnosis of these entities.

Answer: We agree with the reviewer and have now changed the order of the sections, thereby including Pathology under the section Diagnosis and according to the recommendation.

  1. Authors should mention Staging (the procedures for staging are approached in the Diagnosis and Treatment sections).

Answer: Staging has been mentioned, including reference.

Reviewer 2 Report

Comments and Suggestions for Authors

Dear Editor and Authors,

It was my pleasure to evaluate this manuscript titled “Lung Carcinoids: A Comprehensive Review for Clinicians” by Dr. Granberg and colleagues from the Karolinska University Hospital in Stockholm, Sweden.

In this review article the authors attempt to review and present the pathology, the symptoms, the diagnosis and the treatment of patients with lung carcinoids.

This is a very thorough and extensive review which is clearly presented and well written and illustrated. The language was easily understood with very minor grammatical/syntax mistakes and only needs some minor proofreading prior to publication. Overall it was a pleasure to read this review as it provided me, a thoracic surgeon, with a very nice and concise overview of the pathology, presentation and management. I therefore only have some minor comments:

1.       In the surgical treatment section I would suggest the authors elaborate more in terms of open thoracotomy versus VATS/thoracoscopic surgery. The pro and cons of each technique. Also, sleeve lobectomy versus bronchotomy for intraluminal masses, description how it is performed, benefits of each, ect. Basically since surgery is such a big component of the treatment of Carcinoid tumours the authors need to expand the section to include more literature on the subject, describe more the techniques surgeon use, describe peri-operative and long term outcomes and so forth!!

2.       Again in the surgery section the whole laser removal ect should be moved in a different section possibly under interventional pulmonology! Overall, the surgical section as mentioned above is quite short and incomplete and needs significant expansion!!

3.       I am not sure I understand what the whole section about “Local Treatment” is about? Do they authors mean liver metastasis management? This section needs some more elaboration and clarification what it pertains to!

In conclusion, as mentioned above this is an excellent report which only needs some minor additions to make it more complete and ready for publication. I commend the authors for their work. Kind regards to all.

Comments on the Quality of English Language

Minor language proofreading is needed. Not much.

Author Response

Response to Reviewer 2:

  1. In the surgical treatment section I would suggest the authors elaborate more in terms of open thoracotomy versus VATS/thoracoscopic surgery. The pro and cons of each technique. Also, sleeve lobectomy versus bronchotomy for intraluminal masses, description how it is performed, benefits of each, ect. Basically since surgery is such a big component of the treatment of Carcinoid tumours the authors need to expand the section to include more literature on the subject, describe more the techniques surgeon use, describe peri-operative and long term outcomes and so forth!!

Answer: Thank you for this valid comment. The section about surgery has now been expanded, references have been added, and VATS has also been included.

  1. Again in the surgery section the whole laser removal ect should be moved in a different section possibly under interventional pulmonology! Overall, the surgical section as mentioned above is quite short and incomplete and needs significant expansion!!

Answer: The section about laser treatment has been moved to a new section mentioned “Interventional pulmonology” and expanded according to the suggestion from the reviewer. 

  1. I am not sure I understand what the whole section about “Local Treatment” is about? Do they authors mean liver metastasis management? This section needs some more elaboration and clarification what it pertains to!

Answer: We have now tried to elaborate this section and clarify the methods mentioned, as suggested by the reviewer.
